# Genome-Wide Identification and Expression Analysis of *TGA* Family Genes Associated with Abiotic Stress in Sunflowers (*Helianthus annuus* L.)

**DOI:** 10.3390/ijms25074097

**Published:** 2024-04-07

**Authors:** Qinzong Zeng, Jiafeng Gu, Maohong Cai, Yingwei Wang, Qinyu Xie, Yuliang Han, Siqi Zhang, Lingyue Lu, Youheng Chen, Youling Zeng, Tao Chen

**Affiliations:** 1Xinjiang Key Laboratory of Biological Resources and Genetic Engineering, College of Life Science and Technology, Xinjiang University, Urumqi 830017, China; zengqz@stu.xju.edu.cn; 2College of Life and Environmental Science, Hangzhou Normal University, Hangzhou 311121, China; gujiafeng2023111010061@stu.hznu.edu.cn (J.G.); caimaohong@hznu.edu.cn (M.C.); wangyingwei@stu.hznu.edu.cn (Y.W.); xieqinyu@stu.hznu.edu.cn (Q.X.); hanyuliang@stu.hznu.edu.cn (Y.H.); 2022011010002@stu.hznu.edu.cn (S.Z.); 2022111010025@stu.hznu.edu.cn (L.L.); 18358172013@163.com (Y.C.)

**Keywords:** *Helianthus annuus* L., transcription factor, TGA, abiotic stress, salt

## Abstract

Sunflower (*Helianthus annuus* L.) is an important, substantial global oil crop with robust resilience to drought and salt stresses. The TGA (TGACG motif-binding factor) transcription factors, belonging to the basic region leucine zipper (bZIP) family, have been implicated in orchestrating multiple biological processes. Despite their functional significance, a comprehensive investigation of the TGA family’s abiotic stress tolerance in sunflowers remains elusive. In the present study, we identified 14 TGA proteins in the sunflower genome, which were unequally distributed across 17 chromosomes. Employing phylogenetic analysis encompassing 149 TGA members among 13 distinct species, we revealed the evolutionary conservation of TGA proteins across the plant kingdom. Collinearity analysis suggested that both *HaTGA01* and *HaTGA03* were generated due to *HaTGA08* gene duplication. Notably, qRT-PCR analysis demonstrated that *HaTGA04*, *HaTGA05*, and *HaTGA14* genes were remarkably upregulated under ABA, MeJA, and salt treatments, whereas *HaTGA03*, *HaTGA06*, and *HaTGA07* were significantly repressed. This study contributes valuable perspectives on the potential roles of the *HaTGA* gene family under various stress conditions in sunflowers, thereby enhancing our understanding of *TGA* gene family dynamics and function within this agriculturally significant species.

## 1. Introduction

Transcription factors (TF) can specifically regulate the expression of plant genes, serving as pivotal enhancers of the plant’s capacity to adapt to stressful environments [1]. Among these regulatory proteins, the basic leucine zipper (bZIP) gene family is distinguished by its crucial role in plant development, stress response, and hormone synthesis [2,3]. Based on the highly conserved domains and motifs, bZIP genes have been categorized into 10 primary groups (A, B, C, D, E, F, G, H, I, and S). In addition, there are two extra groups named J and K, further expanding the bZIP gene family diversity within the plant kingdom [4,5]. The TGA TFs are a part of group D, which is crucial for both plant development and pathogen defense mechanisms by specifically recognizing and binding the TGACG motif of their target genes [6]. Actually, the first TGA member, named TGA1a, was identified in *Nicotiana tabacum* by using the TGACG motif to screen the cDNA library [7]. Typically, a canonical TGA transcription factor consists of a highly variable N region, bZIP domain, and C region. The N-terminal region of TGAs, with the conserved phosphorylation motif STDxDT in it, is postulated to be instrumental in conferring functional specificity [8]. The bZIP domain, which has a leucine zipper structure for dimerization, is required for specific target promoter binding. Moreover, a nuclear localization signal has been identified in this domain [9]. The C-terminal region of TGAs consists of 250 residues approximately, with low variability in all TGA proteins. It is worth noting that the DOG1 (Delay of Germination 1) domain is present in this region. Since DOG1 is identified as a short TF-like protein without DNA binding ability, it is assumed that the C-terminal region of TGAs may participate in the orchestration of TGA activity [10].

Research in strawberries has identified several TGA members, specifically *FaTGA1/2/5/7/8/9/10*, as participants in the defense against diseases such as powdery mildew [11]. In Arabidopsis, TGA2/5/6 factors have been implicated in the regulation of the antioxidant and detoxification pathways in response to UV-B exposure [12]. Additionally, in *Solanum lycopersicum*, brassinoidsteroid (BR) induces the expression of TGA2, which in turn triggers the removal of pesticide residues in tomatoes by inducing apoplastic ROS [13]. The TGA factors have been associated with enhanced resistance to biotic stressors in various plant species; for instance, the *StTGA* gene has been linked to the improved ability to withstand bacterial wilt in *Solanum tuberosum* (potato) [14]. *GmTGAs* have been reported to improve symbiotic nodulation and nitrogen use efficiency in *Glycine max* (soybean) [15]. Furthermore, the pervasive influence of TGA proteins on plant abiotic stress responses is underscored by studies. Overexpressing *GmTGA17* in Arabidopsis and soybeans boosts plants’ stress tolerance [1]. The *MhTGA2* gene was induced by salt, drought, and low-temperature stresses in *Malus hupehensis* [16]. In addition to functional roles in stress response pathways, *TGA1* and *TGA4* have been recognized as important regulatory factors of the nitrate response in Arabidopsis roots, exemplifying the versatility of these factors in nutrient signaling and utilization [17]. Currently, the *TGA* gene family has been identified and functionally analyzed in soybeans [1], peanuts [18], and potatoes [14], where 27, 20, and 42 *TGA* genes have been identified, respectively. The ongoing elucidation of *TGA* gene functions continues to reveal new aspects of their contribution to plant resilience and offers promising targets for crop improvement strategies.

Sunflower (*Helianthus annuus* L.) represents one of the four major oilseed crops globally, originating from eastern North America and now extensively cultivated across the world [19]. Adapted to the warm climes of tropical and subtropical regions, sunflowers exhibit remarkable resistance to saline–alkali stress [20,21]. Given their pronounced stress tolerance, sunflowers serve as models for studying abiotic stress responses [22,23]. Since the whole genome sequence of *Helianthus annuus* was reported in 2017 [24], a wide array of TFs have been identified for their roles in responding to abiotic stresses. These encompass the AP2/ERF [25], MYB [26], bHLH [27], NAC [28], and TCP [29] families of TFs. Despite the fact that the *TGA* gene family plays a crucial role in plant development and stress response, there has been no report of TGA TF in *Helianthus annuus* thus far.

In the present investigation, we have described the phylogenetic relationship, chromosomal distribution, gene colinearity analysis, and characteristics of 14 members of the *HaTGA* gene family. Through the employment of qRT-PCR assays, we have uncovered that the expression of the *HaTGA01*, *HaTGA05*, and *HaTGA14* genes is induced by hormones (ABA and MeJA) and abiotic stimuli (anaerobic and salt). These findings provide crucial information about *HaTGA* gene evolution and potential functions, which can guide future functional characterization studies of these genes.

## 2. Results

### 2.1. Identification and Analysis of TGA Family Genes in Sunflowers

In this study, we aimed to catalog members of the TGA transcription factor family within the *Helianthus annuus* (sunflower) genome. To this end, the full-length proteins and conserved domains of well-characterized Arabidopsis TGA proteins were used as BLAST query sequences against the sunflower genome databases. After the elimination of redundant sequences, a total of 14 non-redundant *TGA* genes were identified and designated as *HaTGA01* to *HaTGA14* by their sequential chromosomal arrangement. The distribution of these 14 *TGA* genes in sunflowers was found to be disproportionate across the 17 sunflower chromosomes. Chromosomes 2 and 13 featured the highest density of *TGA* genes, whereas chromosomes 1, 3, 7, 8, and 16 were devoid of any members of this gene family (Figure 1A). To explore the diversity in gene architecture, we analyzed the structural composition of the *HaTGA* genes, including untranslated regions (UTR), exons, and introns, and found that the number of exons in all *HaTGA* genes ranged from 7 to 12, and the number of introns varied from 7 to 13, exhibiting significant variation. Among these TGAs, *HaTGA02* was noteworthy for its complexity, embodying the largest number of exons and introns, with 13 exons and 12 introns (Figure 1B). In addition, to gain a deeper understanding of the structure and evolution of the genome and, thus, to infer the functions and interrelationships of genes, we performed a collinearity analysis of *TGA* genes in sunflowers. The presence of a higher number of exons in *HaTGAs* may indicate a more complex gene structure, possibly due to alternative splicing or gene duplication events. As shown in Figure 1C, our analysis delineated four syntenic gene pairs among the *HaTGA* family, including *HaTGA01*-*HaTGA08*, *HaTGA02*-*HaTGA11*, *HaTGA03*-*HaTGA08*, and *HaTGA06*-*HaTGA13*, which was possibly caused by segment duplication and tandem duplication events. We further aligned their amino acid sequences and found that the protein sequences were highly conserved between each respective pair (Figure 1B and Appendix A), an indication of the molecular stability imparted by chromosomal duplications throughout evolutionary history.

To further characterize the *TGA* gene family, the basic information about *TGA* family genes is shown in Table 1, including amino acids, molecular weight, isoelectric points, instability index, lipid solubility index, and total mean hydrophilicity. The HaTGA protein length ranges from 326 amino acids (HaTGA09) to 512 amino acids (HaTGA11), with the corresponding molecular weight varying from 36.40 kDa (HaTGA12) to 56.91 kDa (HaTGA11). The isoelectric point (PI) ranges from 5.7 (HaTGA13) to 8.92 (TGA08). Analysis of the instability index showed that all *HaTGA* genes were unstable proteins, which might have implications for their functional lifespan in the cell. As predicted by subcellular localization, HaTGAs TF were all predominantly localized to the nucleus. The Aliphatic Index revealed that HaTGAs are hydrophilic proteins. Collectively, this information provides a comprehensive profile of the biophysical and biochemical properties of the HaTGA proteins.

### 2.2. Phylogenetic Analysis of TGA Family Genes

To elucidate the attributes of the TGA transcription factor family in crops, we identified 149 TGA proteins from the model organism *Arabidopsis thaliana* and 12 major crop species. The distribution of these TGA proteins in different species is as follows: *Helianthus annuus* (sunflower) (14), *Arabidopsis thaliana* (10), *Phaseolus vulgaris* (common bean) (8), *Arachis hypogaea* (peanut) (20), *Glycine max* (soybean) (27), *Oryza sativa* (rice) (15), *Lactuca sativa* (lettuce) (10), *Sesamum indicum* (sesame) (9), *Nicotiana tabacum* (tobacco) (12), *Sorghum bicolor* (sorghum) (4), *Zea mays* (corn) (5), *Cicer arietinum* (chickpea) (8), and *Vitis vinifera* (grape) (7). Bioinformatics profiling of the 149 *TGA* genes showed that the protein length and molecular weight of HaTGA ranged from 290 (GmTGA05) to 540 aa (OsTGA07) and 5.7 (HaTGA13) to 8.92 (HaTGA08), respectively. The evaluation of the hydrophilicity profiles revealed that except for AtTGA01 and VvTGA05, the other 147 TGA proteins are unstable, with an aliphatic index ranging from 68.73 (AhTGA01) to 97.81 (AhTGA05). The hydrophilic results indicated that TGA proteins are generally hydrophilic in nature (Appendix A).

To evaluate the evolutionary development of *TGA* gene families, multiple sequence alignments and the construction of phylogenetic trees of the 149 TGA proteins isolated from 13 species were conducted. The phylogenetic analysis indicated that the TGA proteins were categorized into four principal clades: Group I, Group II, Group III, and Group IV. Each group comprised 149, 71, 41, 20, and 17 TGA proteins, respectively, rendering a comprehensive overview of the family’s diversification (Figure 2A). Within this classification, HaTGA proteins were found to be distributed across all four clades. Specifically, Group I to Group IV contained five, six, two, and one HaTGA(s), respectively. Notably, sorghum and corn TGA proteins were exclusively present within Group I, highlighting a potentially conserved evolutionary trace in these cereals (Figure 2B).

### 2.3. Conserved Motif and Domain Analysis of the TGA Family Genes

Motifs, comprising conserved sequences with biological functions, usually determine protein specificity and function through their distinctive motif compositions. To further analyze the functional domains within the 149 TGA family members, the motif composition of these TGA family members was analyzed by utilizing the Multiple Expectation Maximization for Motif Elicitation (MEME) suite. This analysis successfully identified a repertoire of ten conserved motifs among these TGA proteins, with the length of the motifs ranging from 15 to 50 amino acids. The relative height of the letter designating the amino acid residue at each position represented the degree of conservation. A comprehensive depiction of these motifs, along with the conserved domains present across all TGA members (Appendix A), is shown.

Although there are overarching similarities in the number and location of motifs, a distinct divergence in motif composition is discernible across the four groups of TGA (groups I, II, III, and IV). An analysis of motif distribution revealed that the predominant number of *TGA* genes across all groups shared motifs 1–7. Specifically, in Group I, a majority of the TGAs consist of nine motifs, except motif 9. The composition of TGAs in Group II primarily includes motifs 1–7, and 9. Group III is predominantly comprised of motifs 1–7, whereas Group IV is mainly constituted by motifs 1–8. These variations in motif patterns suggest a significant relationship between the differences in motif composition and the functional differences of TGA proteins within the four different groups (Figure 3B). Further domain analysis revealed that all members of the TGA family possess a Delay of Germination 1 (DOG1) domain and a basic leucine zipper (bZIP) domain, with motifs 3 and 7 constituting the bZIP domain and motifs 1 and 5 mainly contained in the DOG1 domain (Figure 3C).

Meanwhile, to gain insight into the structural implications of these motifs and potentially elucidate corresponding secondary structures, the AlphaFold (https://colab.research.google.com/github/sokrypton/ColabFold/blob/main/AlphaFold2.ipynb, accessed on 2 February 2024) computational tool was employed to predict the protein structure of HaTGA. Accordingly, the 3D structure of HaTGA01 in Appendix A includes all three fundamental structural components integral to sunflower TGA TFs. These encompass the conserved bZIP domain, the highly variable N-terminal region (colored orange), and the C-terminal portion (C-terminal alpha helix), which contains a hypothetical Delay of Germination 1 (DOG1) domain [30].

### 2.4. Distribution and Function of Cis-Acting Elements in the TGA Promoters

The analysis of promoters is essential to understanding the regulation of transcription and the potential functions of *HaTGA* genes. To investigate the transcriptional regulation of *TGA* genes, the 2000 bp sequences that flank the upstream region of *TGA* family genes from 13 species were extracted and subjected to cis-element analysis via the PlantCARE database. We have found a total of 10 cis-regulatory elements related to ABA response, MeJA response, drought tolerance, GA response, and low-temperature adaptability (Figure 4A). Further functional analysis indicated that the ABA-responsive elements, MeJA-responsive elements, and anaerobic-induction elements were abundant in the promoter of these *TGA* family genes (Figure 4B). Notably, the presence of 10 distinct ABRE motifs in the ABA response and two unique motifs, namely the TGACG-motif and CGTCA-motif, in MeJA responsiveness highlights the complexity of plant signaling pathways. These findings emphasize the importance of further research in this area to gain a deeper understanding of these mechanisms and their potential applications in agriculture and biotechnology.

### 2.5. Collinearity Analysis of TGA Genes among Plant Species

To understand the phylogenetic affiliations among *TGA* genes of sunflower and other different plant species, including Arabidopsis, peanut, soybean, lettuce, sesame, tobacco, sorghum, corn, chickpea, and grape, a comprehensive collinearity analysis was conducted. As shown in Figure 5, the resulting collinearity assessment revealed that the relationship of *TGA* genes between sunflower and dicotyledonous plants was closer than that with monocotyledonous plants, with soybean being the most closely related.

Quantitatively, there were 22, 19, 18, 7, 7, 6, and 4 syntenic gene pairs linking sunflower with other dicotyledons, including soybean, peanut, lettuce, sesame, chickpea, grape, and Arabidopsis, respectively. Comparatively, the homologous *TGA* gene pairs between sunflower and monocotyledonous species were significantly reduced. Unlike the peanut, sunflowers have been identified with four and one gene pairs with sorghum and maize, respectively. Notably, an exception was observed wherein no direct homologous gene pairs were detected between sunflower and tobacco. This study highlights the evolutionary nuances distinguishing dicot and monocot *TGA* gene sets, with implications for understanding the genomic context of gene family evolution in plant species and spurring future investigations into their functional conservation and divergence.

### 2.6. Expression Profiles and Patterns of HaTGA Genes in Response to Hormonal and Abiotic Stresses

In light of the results obtained from the cis-element analysis and the established connection between the *TGA* gene and stress response mechanisms, the analysis reveals an enriched presence of cis-regulatory elements within the promoters of *HaTGA* genes. Specifically, these promoter regions are replete with sequences associated with key physiological responses, including abscisic acid (ABA) and methyl-jasmonate (MeJA) signal transduction and anaerobic stress adaptability. Such a group of cis-regulatory motifs indicated the *HaTGA* gene promoters in a potentially broad and sophisticated network of regulatory pathways pivotal for plant stress tolerance.

To investigate the expression dynamics of *HaTGA* genes in response to specific abiotic stress, we conducted qRT-PCR analysis to evaluate the expression profiles of the 14 identified *HaTGA* genes. The investigation revealed that *HaTGA01*, *HaTGA04*, *HaTGA05*, and *HaTGA14* exhibited a substantial increase in expression levels in response to ABA and MeJA treatments, whereas *HaTGA03*, *HaTGA06*, *HaTGA07*, and *HaTGA12* demonstrated notable suppression under these conditions (Figure 6 and Figure 7). These observations suggested pivotal roles for these genes in hormone stress responses. Notably, *HaTGA02* and *HaTGA04* displayed a transient expression peak at 3 h of ABA treatment (Figure 6). Concurrently, the expression levels of *HaTGA11* and *13* were significantly down-regulated within 4 h under MeJA treatment (Figure 7). The expression analysis of *HaTGA* under anaerobic stress suggested that, despite *HaTGA05* being induced under these conditions, most *HaTGA* expression was suppressed in sunflowers under anaerobic conditions, indicating their potential role in abiotic stress responses. This suppression led to the speculation that *HaTGA05* might act as a crucial regulatory gene to sustain the survival of the plant in anaerobic environments (Appendix A).

Considering sunflowers as a pioneer crop in saline–alkali soil and *TGA* family genes have been documented to play a vital role in response to salt stress, it is necessary to study the function of *TGA* participating in salt response in sunflowers. To examine how *HaTGA* genes respond to salt stress, we carried out qRT-PCR analysis to assess the expression patterns of the 14 identified *HaTGA* genes. The results suggested that under salt stress, *HaTGA02*, *HaTGA04*, *HaTGA05*, *HaTGA11*, and *HaTGA14* exhibited induced expression at different time points; particularly, the expression of *HaTGA14* increased 5-fold after 6 h of salt stress. Conversely, the *HaTGA03*, *HaTGA06*, *HaTGA07*, *HaTGA08*, *HaTGA10*, *HaTGA12*, and *HaTGA13* genes were markedly down-regulated in response to salt stress. Meanwhile, *HaTGA01* was significantly up-regulated at 3 h and then significantly down-regulated (Figure 8). These data indicate the important functions of *TGA* in response to salinity. This infers the potential utility of leveraging *TGA14* and other *TGA* genes within breeding programs aimed at enhancing sunflower resilience to saline conditions.

## 3. Discussion

The systematic investigation of transcription factor gene families is pivotal for dissecting gene functions and delineating regulatory networks [31]. The *TGA* gene family performs crucial functions in plant development, growth, and defense mechanisms [32,33]. However, their functions in crop stress have been rarely reported. To date, the involvement of *TGA* family genes in sunflowers’ responses to abiotic stress has remained unexplored. In our present study, we have characterized the *HaTGA* genes in sunflowers at the genome-wide level. Our analysis revealed the presence of 14 *HaTGA* genes in sunflowers, all of which contain the canonical bZIP domain with a non-conserved N terminus and a C terminus containing a putative germination delay 1 (DOG1) domain. Our study has elucidated that certain groups of genes, namely *HaTGA01*, *HaTGA03*, *HaTGA08; HaTGA02*, *HaTGA11*; and *HaTGA06*, *HaTGA13* (Figure 1B,C), exhibit close relatedness in the phylogenetic tree and collinearity analysis. We have observed that gene duplication events occur across a wide span of chromosomes rather than being confined to neighboring chromosomes. Furthermore, we have noted that each pair of genes has one gene located at the chromosome terminus, while most genes without duplication events are positioned within the middle of the chromosome. This observation raises the possibility that these loci may be more predisposed to gene duplication and transfer (Figure 1A,C). Notably, branches *HaTGA06*-*13* and *HaTGA08*-*01*-*03* are grouped, indicating a shared evolutionary history. However, within this branch, there is an additional member, member 4. The relationship between members *HaTGA04* and *HaTGA08*-*01*-*03* necessitates further investigation to elucidate their evolutionary connections and potential functional implications (Figure 1B). This suggests that their position on the chromosome may be related to a common regulation or regulatory mechanism. Both tandem and segmental duplications contributed to the diversity of TGA. In abiotic stress responses, we observed that among these four pairs of homologous genes, *HaTGA03*-*HaTGA08* and *HaTGA06*-*HaTGA13* consistently exhibit a downregulation trend, regardless of hormone treatment, salt stress, or anaerobic stress. Meanwhile, *HaTGA02*-*HaTGA11* shows a similar upregulation trend only under salt stress. However, a different pattern is observed for the homologous gene pairs *HaTGA01* and *HaTGA08*. Apart from exhibiting consistent downregulation under anaerobic stress, they display opposite expression trends in other treatments. For example, under ABA and MeJA treatments, *HaTGA01* is significantly upregulated, whereas *HaTGA08* is markedly downregulated. This indicates that these homologous genes may have retained similar functions during evolution or are regulated by similar mechanisms. This consistency may indicate that these genes have important shared roles in sunflower’s response to abiotic stress, potentially participating in similar biological processes or pathways (Figure 6, Figure 7, Figure 8 and Appendix A).

Furthermore, we also identified a total of *149 TGA* genes from ten dicots and two monocots for phylogenetic analysis. Based on sequence characteristics and phylogenetic analysis, we classified these *TGA* genes into four groups (Figure 2A). The significantly higher numbers of members in Group I and Group II compared to Group III and Group IV suggest that Group III and Group IV likely originated earlier in evolutionary history. Both monocots and dicots were distributed in the first group. This indicates that TGA proteins diverged with plant evolution. We further analyzed their motifs and conserved domains (Figure 3B,C and Appendix A) and found a high degree of structural similarity. Gene duplication serves as a pivotal mechanism driving the formation of gene families [34]. Therefore, it is postulated that the aforementioned seven *HaTGA* genes may have originated from chromosomal duplication events during plant evolution.

The genome collinearity analysis of *TGA* genes between sunflowers and other plants reveals several significant patterns. Firstly, in dicotyledonous plants, sunflower’s *HaTGA02*, *07*, *11*, and *14* exhibit extensive collinearity with most dicotyledonous plants (Figure 5A–G), whereas in monocotyledonous plants, only *HaTGA02*, *07*, and *11* show collinearity with monocotyledonous *TGA* genes (Figure 5H,I and Appendix A). This suggests that these TGA proteins may serve as the evolutionary origins of the *TGA* gene family. Moreover, in conjunction with Sunflower’s collinearity analysis, the earliest produced TGA proteins are inferred to be 02 and 07, wherein 02 protein duplicates to generate 11 proteins, forming Group III, while 07 protein evolves independently to form another branch (Group IV). Lastly, based on the presence or absence of conserved motif 8 in the protein analysis (Figure 3B), it is speculated that Group II evolves from Group III, while Group I evolves from Group IV.

Phylogenetic analysis has demarcated a cluster designated as Group I, within which a body of prior research has provided functional insights. Specifically, the ectopic overexpression of *AtTGA05* has been associated with SAR-independent resistance against the pathogenic oomycete *Peronospora parasitica* in *Arabidopsis thaliana* [35]. *OsTGA02* enhances rice disease resistance to leaf blight, suggesting its primary role in defense signaling pathways in *Oryza sativa* [36]. Based on the phylogenetic affinities and drawing upon functional parallels from closely related species, we hypothesize that five HaTGA TFs (HaTGA14, HaTGA12, HaTGA05, HaTGA09, and HaTGA10), which belong to the same group I, may play a role in conferring resistance to biotic stress.

Apart from sunflower, the promoter elements of the *TGA* gene family in other species are primarily concentrated in ABA, MeJA, and anaerobic stress response elements, with additional types of promoter elements present. This suggests that the functionality of the *TGA* gene family may have diversified in other species. In other words, members of the *TGA* gene family in different species may have evolved distinct functional characteristics, leading to variations in their responses to various environmental stresses and hormones. Therefore, this functional divergence indicates that the *TGA* gene family may have undergone adaptive evolutionary changes across different species (Figure 4B).

Sunflower TGA TFs are predominantly classified within group II, as illustrated in Figure 2A. Our investigation reveals that *HaTGA01*, *HaTGA04, HaTGA03*, *HaTGA06*, and *HaTGA08* exhibit responsiveness to ABA signaling (Figure 6). ABA activation of TFs is crucial for maintaining plant equilibrium under stressful conditions such as dehydration or salt stress [37]. Previous studies have demonstrated that *AtTGA4*, a member of the corresponding phylogenetic group, is activated in response to drought and low nitrogen stress conditions. Remarkably, the overexpression of *AtTGA4* in Arabidopsis has been associated with improved drought tolerance and mitigates nitrogen starvation [38]. Furthermore, overexpression of the *AhTGA11* gene in Arabidopsis enhances resistance to cold and drought stress [18]. Under osmotic stress, plants synthesize ABA, which is crucial for their stress response and tolerance. ABA regulates gene expression in response to osmotic stress and also plays a role in controlling gene expression during seed development and germination. Modulating the expression of ABA signaling factors may enhance plant tolerance to environmental stresses [39]. Based on these findings, our study posits that *HaTGA01*, *HaTGA04, HaTGA03*, *HaTGA06*, and *HaTGA08* may play important roles in enhancing sunflower resilience against salinity and drought stress, potentially via the modulatory functions of the ABA signaling pathway.

Furthermore, within the gene cluster, *HaTGA03*, *HaTGA04*, *HaTGA06*, *HaTGA08*, and *HaTGA13* have been identified to exhibit a dual responsiveness to both MeJA and NaCl (Figure 7 and Figure 8). MeJA is recognized as a phytohormone that orchestrates plant defense against biotic stress [40]. It has also been documented that MeJA improves the salt tolerance of juvenile seedlings by enhancing plant growth, bolstering antioxidant activity, and mitigating excessive Na^+^ accumulation [41]. Plant response to MeJA implies its efficacy against various stressors, including pathogens, salt stress, drought stress, low temperature, heavy metal stress, and other detrimental factors [42]. Furthermore, in connection with abiotic stress and hormonal triggers, three group II genes, *AhTGA04*, *AhTGA14*, and *AhTGA20,* have been identified as key players in responding to abiotic stress and hormonal stimulation, particularly in the jasmonic and salicylic acid pathways [18]. Additionally, *GmTGA15* has been reported to enhance resistance to drought stress in both soybean and Arabidopsis plants [43]. Collectively, these findings suggest that the majority of functionally characterized genes within this group are instrumental in enhancing plant tolerance to adverse environmental conditions, such as salinity and drought, through an intricate response to ABA and MeJA signaling. Interestingly, the aforementioned *HaTGA03*-*HaTGA08* and *HaTGA06*-*HaTGA13* all belong to Group II. They respond to ABA, MEJA, and salt stress and exhibit similar expression patterns as sunflowers. This suggests that genes in this branch play a more critical role in coordinating sunflower defense against abiotic stresses. Despite *HaTGA* genes sharing a homologous evolutionary background and genetic architecture with the genes above, it is imperative to conduct further empirical studies to corroborate the hypothesis that *HaTGA* members of Group II are integrally associated with abiotic stress response mechanisms.

## 4. Materials and Methods

### 4.1. Identification of Members of the TGA Gene Family in Helianthus annuus L.

The *HaTGA* genes were identified using BLAST (https://blast.ncbi.nlm.nih.gov/Blast.cgi, accessed on 15 October 2023) and hidden Markov model (HMM)-based search methods [44]. The sunflower genomic data were retrieved from the EnsemblPlants database (https://plants.ensembl.org/, accessed on 14 October 2023). The sequences of known 10 TGA protein sequences (i.e., *AtTGA01* (AT5G65210), *AtTGA02* (AT5G06950), *AtTGA03* (AT1G22070), *AtTGA04* (AT5G10030), *AtTGA05* (AT5G06960), *AtTGA06* (AT3G12250), *AtTGA07* (AT1G77920), *AtTGA08* (AT1G68640), *AtTGA09* (AT1G08320), and *AtTGA10* (AT5G06839)) from Arabidopsis were retrieved from the TAIR database (https://www.arabidopsis.org/, accessed on 10 October 2023). These sequences were used as queries to search against the sunflower genome files and gene annotation files using the BLASTP (https://blast.ncbi.nlm.nih.gov/Blast.cgi?PROGRAM=blastp&PAGE_TYPE=BlastSearch&LINK_LOC=blasthome, accessed on 25 October 2023) program with a threshold E-value cutoff of 1.0 × 10^−5^ [45] and remove duplicates from the filtered sequence. Obtaining candidate sunflower genes. Subsequently, each prospective sunflower TGA sequence was employed as a query to the Pfam database (http://pfam.xfam.org/, accessed on 25 October 2023) to confirm its membership in the bZIP (basic leucine zipper, accession number: PF00170) family [46] and DOG1 (delay of germination 1, accession number: PF14144) domain. Repeat sequences were removed manually, and the threshold E-value cutoff was 1.0 × 10^−5^. The results of the two methods were compared, and the common sequence ID was selected. All the discovered proteins were submitted to the databases CDD (https://www.ncbi.nlm.nih.gov/Structure/bwrpsb/bwrpsb.cgi, accessed on 5 December 2023) and SMART (https://smart.embl.de/, accessed on 5 December 2023) to check if they contain full bZIP and DOG1 domains. This integrative approach conclusively delineated the members of the *TGA* gene family within sunflowers and ended with the prediction of 14 *HaTGA* genes, which were designated *HaTGA01* through *HaTGA14*. The ExPASy (https://www.expasy.org/, accessed on 10 December 2023) server was used to predict several physio-chemical parameters of TGA proteins, such as molecular weights (Mw) and theoretical isoelectric points (pI) [47]. The detailed characteristics of the identified *HaTGA* genes are provided in Table 1.

### 4.2. Phylogenetic Tree Analysis of TGA Proteins

To further elucidate the phylogenetic relationship between the HaTGA protein and other plant species, a comprehensive phylogenetic tree was constructed. TGA protein from distinct species such as Arabidopsis (*Arabidopsis thaliana*), common bean (*Phaseolus vulgaris*), peanut (*Arachis hypogaea*), soybean (*Glycine max*), rice (*Oryza sativa*), lettuce (*Lactuca sativa*), sesame (*Sesamum indicum*), tobacco (*Nicotiana attenuata*), sorghum (*Sorghum bicolor*), maize (*Zea mays*), chickpea (*Cicer arietinum*), and grape (*Vitis vinifera*) were obtained in the same way as sunflower. Subsequently, the neighbor-joining (NJ) tree was constructed after the TGA protein sequences were aligned by MEGA-11. Tree nodes were evaluated through 1000 repeated bootstrapping analyses [48]. The final version of the phylogenetic tree was visualized using Evolview (https://www.evolgenius.info/, accessed on 5 January 2024) with modifications, facilitating an intuitive interpretation of the evolutionary interrelations among the TGA proteins under study.

### 4.3. Chromosomal Location Density and Gene Collinearity Analysis of TGA Genes

Based on gene annotation information, the bioinformatic software TBtools (V2.080) was used to analyze the chromosome location and density of sunflower *TGA* family genes. The requisite data for this investigation were downloaded from the Ensembl Plants database. Pair-wise all-against-all BLAST analysis was performed for sunflower and other plant species protein sequences. The results generated from this BLAST analysis, along with the gff3 annotation files, were then determined by the Multiple Collinearity Scan toolkit (MCScanX, V2.3.1) to discern syntenic relationships and potential collinear gene arrangements within the analyzed genomes.

### 4.4. Conserved Domains and Gene Structure Analysis of TGA Genes in Sunflower

To analyze the protein domain structure of TGA family proteins, the SMART (https://smart.embl.de/, accessed on 5 December 2023) online tool was utilized (https://smart.embl.de/), and a functional domain diagram was created using the Conserver Domain Database (CDD) from NCBI (https://www.ncbi.nlm.nig.gov/cdd/, accessed on 5 December 2023). The conserved motif within the candidate *HaTGA* gene sequences was predicted using the MEME Suite 5.5.5 (https://meme-suite.org/, accessed on 25 December 2023) with default parameters. The resulting conserved motifs were visualized, and the gene structure diagrams were generated using the Gene Structure View program in TBtools.

### 4.5. Cis-Regulatory Elements Analysis in TGA Promoters

To elucidate the cis-regulatory elements of *TGA* family genes, the promoter sequences were used to predict cis-acting elements by PlantCARE (https://bioinformatics.psb.ugent.be/webtools/plantcare/html/, accessed on 27 December 2023). The *HaTGA* gene promoter sequences were visually represented using the Simple Biosequence Viewer (V2.080) program in TBtools based on the sunflower promoter sequence files.

### 4.6. Plant Material and Treatments

To investigate the transcriptional response of the *TGA* family genes to abiotic stress, *Helianthus annuus* cultivar AZB was utilized as the experimental material. Sunflower seeds were placed between layers of paper and moistened with sterile water. The paper beds containing the seeds were then placed in Petri dishes and cultured at 25 °C under a photoperiod of 16 h of light and 8 h of darkness for three days to promote germination. After germination, the seedlings were transferred to hydroponic containers and cultured with 1/5 Hoagland’s nutrient solution.

Following two weeks of growth, the seedlings were subjected to various stress treatments by placing them in environments supplemented with 50 µM ABA, 200 µM MeJA, and 300 mM NaCl, respectively. For ABA treatment, leaf samples were collected at 3-h intervals for 3 h, 6 h, and 9 h. MeJA-treated samples were collected at 2-h intervals for 2 h, 4 h, and 6 h after treatment, while NaCl-treated samples were collected at 3-hour intervals for 3 h, 6 h, and 9 h post-treatment. Additionally, anaerobic stress treatment was conducted by submerging two-week-old AZB seedlings in water for 24 h before sampling. Untreated plants served as controls. Given the significance of leaves in photosynthesis and transpiration, three biological replicates of leaf samples were collected for each stress treatment. Post-treatment, leaves were collected for RNA extraction and subsequent qRT-PCR analysis.

### 4.7. RNA Extraction, cDNA Synthesis, and Quantitative Real-Time PCR

Total RNA was extracted from the aforementioned plant material using Trizol reagent (Coolaber, Beijing, China). Moreover, 1 μg of total RNA was used to synthesize cDNA using HiScript II Q Select RT SuperMix for qRT-PCR (Vazyme, Nanjing, China). The qRT-PCR assays were subsequently performed using a CFX384 Touch Real-Time PCR Detection System (Bio-Rad, Hercules, CA, USA) with a reaction volume of 10 μL. The qRT-PCR master mix used was ChamQ universal SYBR qPCR Master Mix (Vazyme, Nanjing, China). Three biological replicate samples were taken for sunflower RNA extraction, and these experiments were independently repeated three times. The *HaTublin* gene was used as a reference gene. The primers used for qPCR are listed in Appendix A.

## 5. Conclusions

In this study, we identified and characterized 14 *HaTGA* genes in sunflowers, including chromosomal distribution, phylogenetic evolution, physicochemical properties, subcellular localization, gene and protein structures, and specific expression patterns. The transcript levels of *HaTGA04*, *HaTGA05*, and *HaTGA14* were upregulated by ABA, MeJA, and salt treatments, whereas *HaTGA03*, *HaTGA06*, and *HaTGA07* were downregulated, consistent with promoter analysis of *HaTGA*s. Furthermore, anaerobic stress significantly inhibited *HaTGA* expression, except for *HaTGA05*. These findings implied their pivotal role in response to biotic and abiotic stresses.

## Figures and Tables

**Figure 1 ijms-25-04097-f001:**
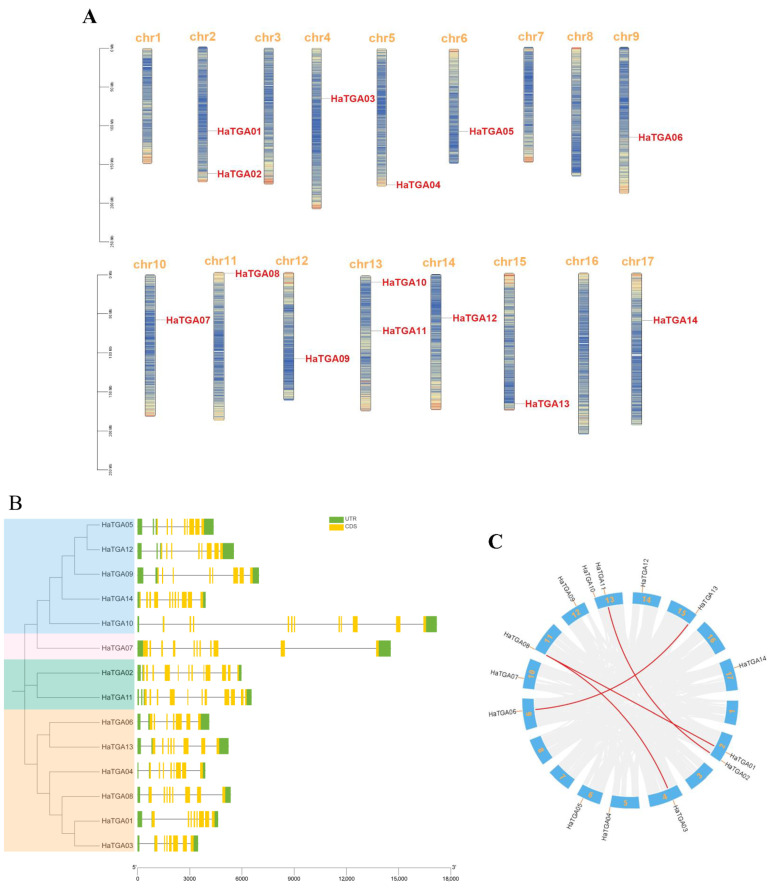
Identification and characterization of *TGA* family genes in sunflower. (**A**) Chromosomal localization of *HaTGA* genes. The gene ID of *HaTGA* is highlighted in red for the corresponding chromosome. The proportional lengths of the chromosomes are represented, with a scale in Mb provided on the left. (**B**) Phylogenetic context and gene architecture of the sunflower *TGA* family. A phylogenic tree delineates the evolutionary relationships among the members, accompanied by exon-intron structure diagrams displaying the untranslated regions (UTRs) and coding sequences (CDS) of the 14 identified sunflower *TGA* genes. (**C**) Syntenic analysis of *H. annuus TGA* genes. The distribution of the 14 *TGA* genes across the sunflower chromosomes is indicated, and syntenic relationships between gene pairs are depicted through connecting red lines.

**Figure 2 ijms-25-04097-f002:**
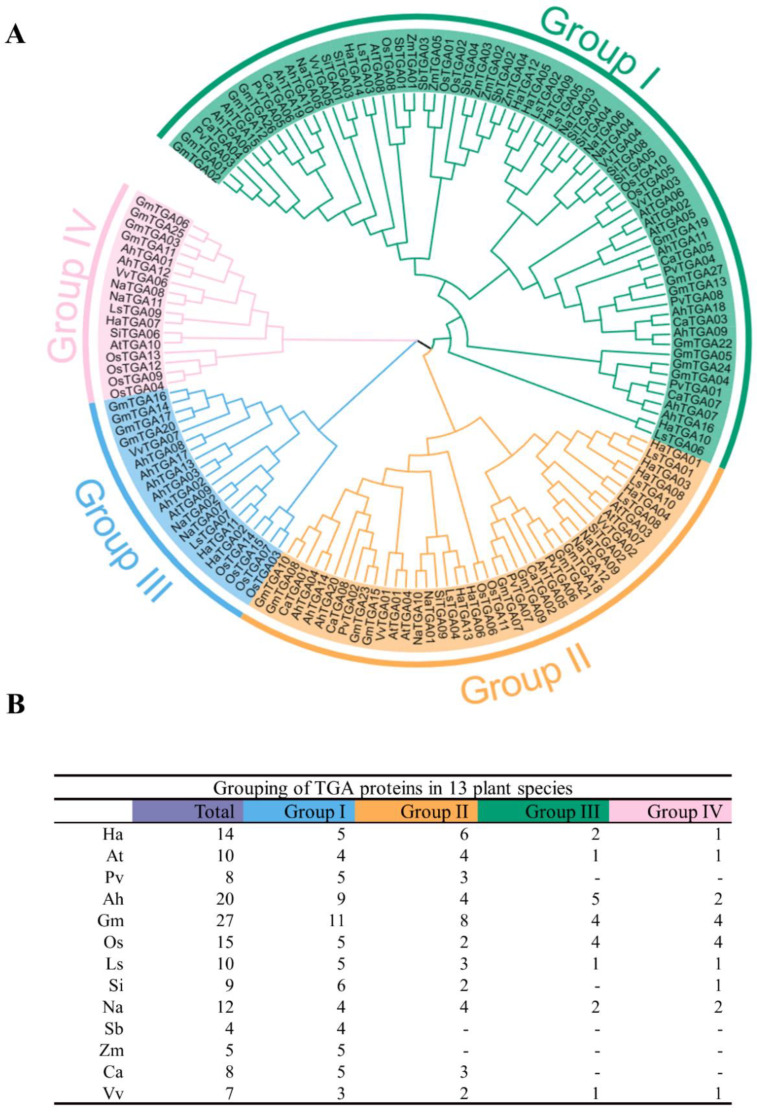
Phylogenetic analysis and classification of TGA proteins across plant species. (**A**) The neighbor-joining phylogenetic tree was constructed using 149 TGA proteins derived from 13 plant species, including *Helianthus annuus* (Ha), *Arabidopsis thaliana* (At), *Phaseolus vulgaris* (Pv), *Arachis hypogaea* (Ah), *Glycine max* (Gm), *Oryza sativa* (Os), *Lactuca sativa* (Ls), *Sesamum indicum* (Si), *Nicotiana attenuata* (Na), *Sorghum bicolor* (Sb), *Zea mays* (Zm), *Cicer arietinum* (Ca), and *Vitis vinifera* (Vv). TGA proteins have been sorted into four clades, highlighted with background colors. (**B**) Taxonomic distribution of TGA proteins among the analyzed plant species. The TGA proteins were categorized according to plant species.

**Figure 3 ijms-25-04097-f003:**
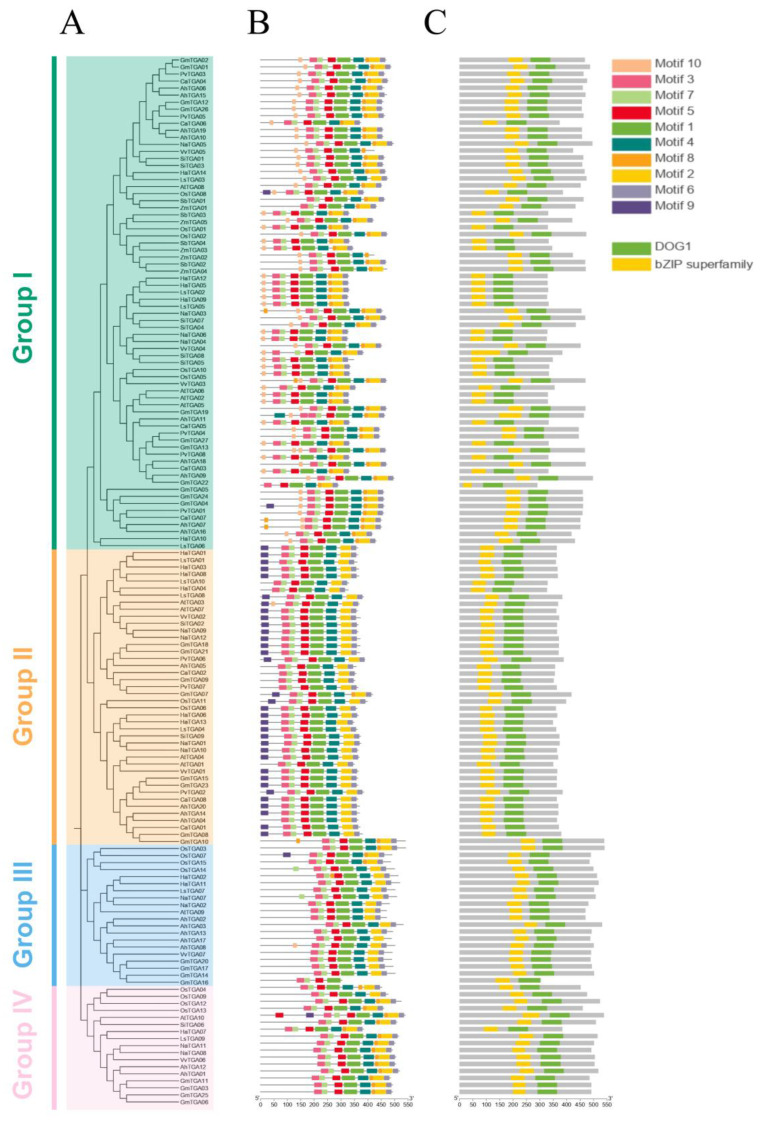
Identification of conserved motifs and protein domains in the *TGA* gene family. (**A**) Phylogenetic reconstruction of the *TGA* gene family. Employing MEGA-11, the phylogenetic tree was generated via the neighbor-joining (NJ) method, with sequences represented by four color codes, delineating the *TGA* genes across plant species. (**B**) Motif composition analysis of the TGA protein. Ten conservative motifs of TGAs were identified by using the MEME. (**C**) Domain architecture of TGA proteins. Two significant protein domains were identified using the NCBI conserved domain database (CDD) and displayed in different colors.

**Figure 4 ijms-25-04097-f004:**
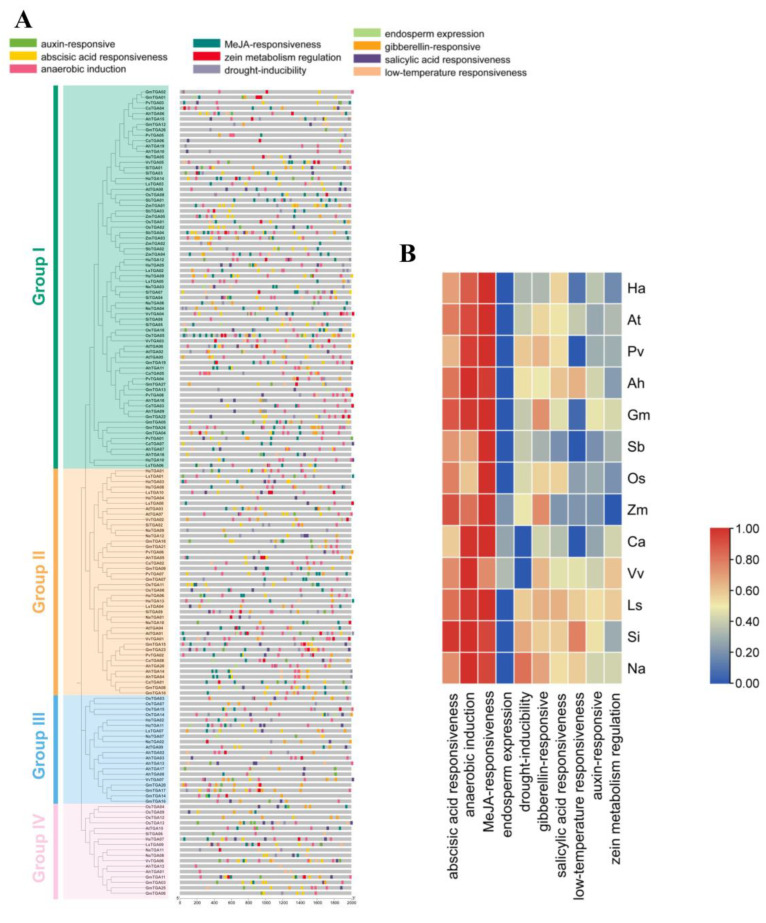
Characterization of the cis-acting elements in the promoter region of *TGA* genes. (**A**) The spatial distribution of the cis-acting motifs. The localization of various cis-acting elements within the promoter regions of *TGA* genes was visualized, with each type of regulatory motif differentiated by a unique color. (**B**) Comparative analysis of cis-elements of *TGA* gene promoters. Cis-regulatory elements residing in the promoters of *TGA* genes from 13 plant species were compiled, categorized, and presented in a heatmap format. The species analyzed include *Helianthus annuus* (Ha), *Arabidopsis thaliana* (At), *Phaseolus vulgaris* (Pv), *Arachis hypogaea* (Ah), *Glycine max* (Gm), *Oryza sativa* (Os), *Lactuca sativa* (Ls), *Sesamum indicum* (Si), *Nicotiana attenuate* (Na), *Sorghum bicolo*r (Sb), *Zea mays* (Zm), *Cicer arietinum* (Ca), and *Vitis vinifera* (Vv).

**Figure 5 ijms-25-04097-f005:**
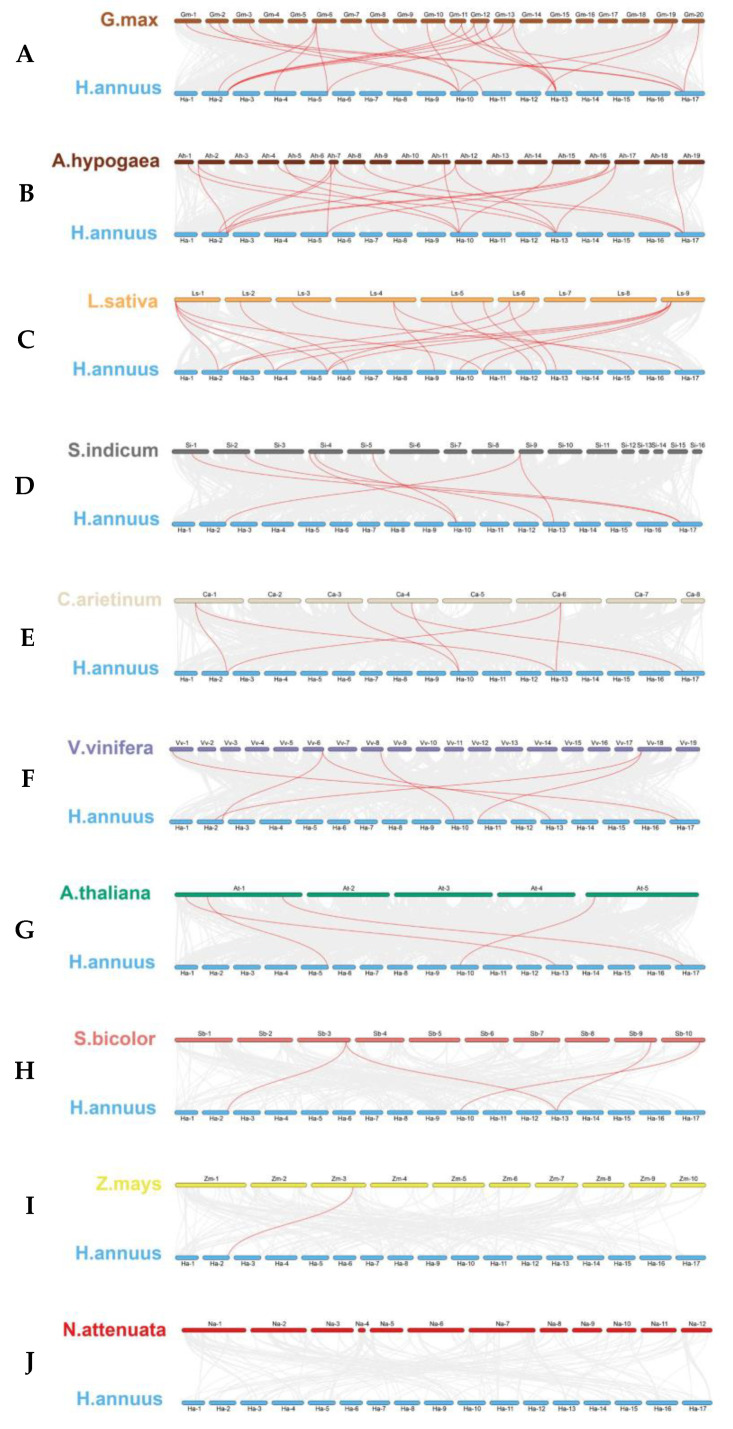
(**A**) *Glycine max* (soybean), (**B**) *Arachis hypogaea* (peanut), (**C**) *Lactuca sativa* (lettuce), (**D**) *Sesamum indicum* (sesame), (**E**) *Cicer arietinum* (chickpea), (**F**) *Vitis vinifera* (grape), (**G**) *Arabidopsis thaliana* (Arabidopsis), (**H**) *Sorghum bicolor* (sorghum), (**I**) *Zea mays* (corn), and (**J**) *Nicotiana attenuata* (tobacco). Chromosomes are distinctly colored, respectively. Gray lines in the background indicate the collinear blocks within *H. annuus* and other plant genomes. Highlighted in red are lines that connect syntenic *TGA* gene pairs.

**Figure 6 ijms-25-04097-f006:**
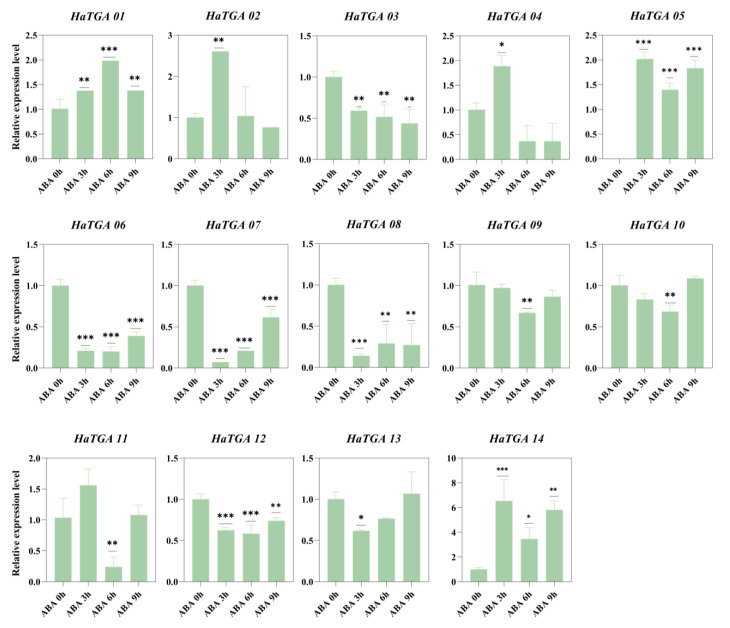
Differential expression profiles of *HaTGA* genes in response to ABA treatment. The expression patterns of *HaTGA* genes in *Helianthus annuus* following exposure to exogenous ABA. The relative expression levels were quantified using the 2^−ΔΔCt^ method with three replicates, normalized against the *HaTubulin* internal control. The values represent the means and standard deviations obtained from three biological replicates. The asterisks indicate statistical significance (* *p* < 0.05, ** *p* < 0.01, and *** *p* < 0.001) compared to the corresponding ABA 0 h.

**Figure 7 ijms-25-04097-f007:**
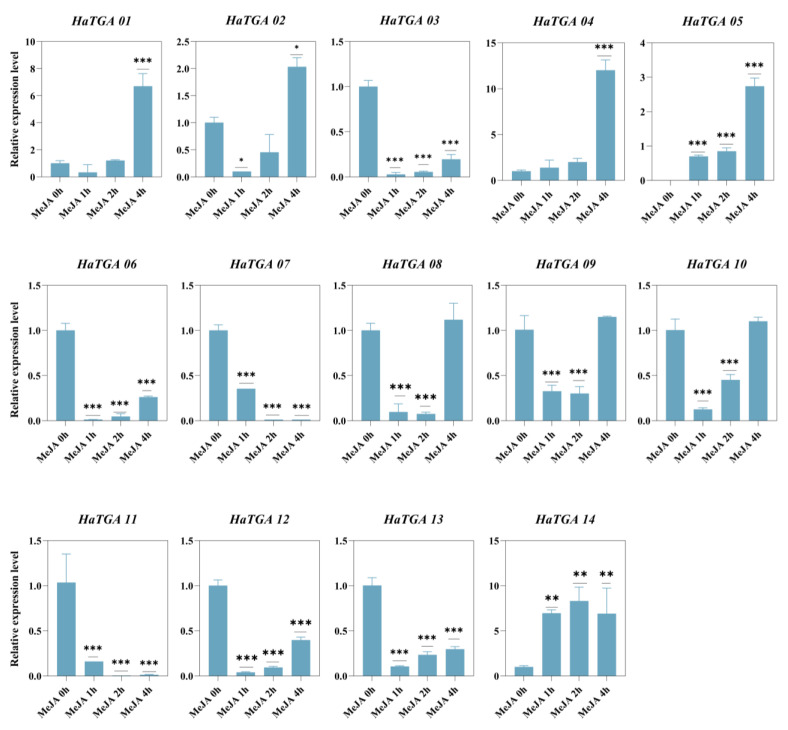
Differential expression profiles of *HaTGA* genes in response to MeJA treatment. The relative expression levels were calculated using the 2^−ΔΔCt^ method with three replicates, normalized against the *HaTubulin* internal control. The values represent the means and standard deviations obtained from three biological replicates. The asterisks indicate statistical significance (* *p* < 0.05, ** *p* < 0.01, and *** *p* < 0.001) compared to the corresponding MeJA 0 h.

**Figure 8 ijms-25-04097-f008:**
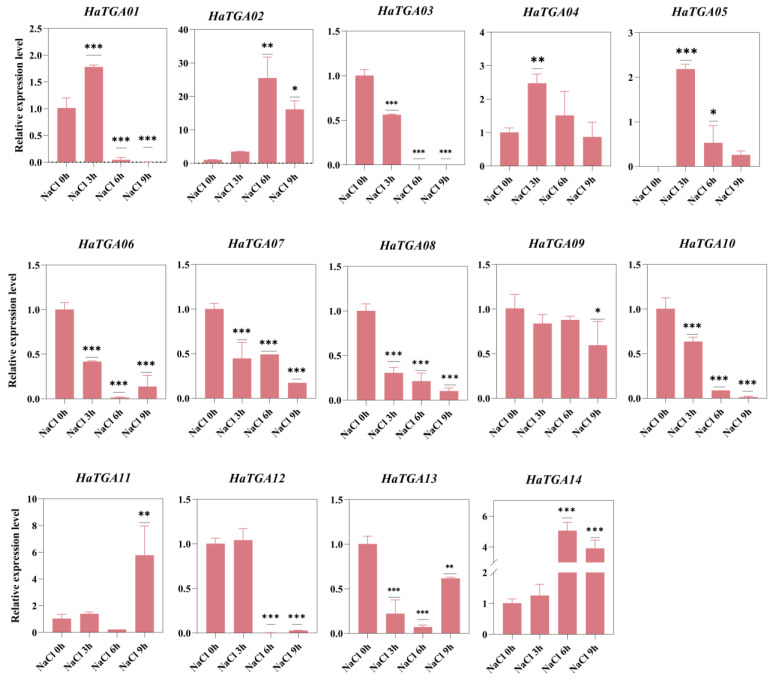
Differential expression profiles of *HaTGA* genes in response to salt stress. The relative expression levels were quantified using the 2^−ΔΔCt^ method with three replicates, normalized against the *HaTubulin* internal control. The values represent the means and standard deviations obtained from three biological replicates. The asterisks indicate statistical significance (* *p* < 0.05, ** *p* < 0.01, and *** *p* < 0.001) compared to the corresponding NaCl at 0 h.

**Table 1 ijms-25-04097-t001:** Basic physical and chemical properties of TGA proteins in sunflower.

Gene ID	Rename	Number of Amion Acids	MW (Da)	PI	Instability Index	Aliphatic Index	Grand Average of Hydropathicity	Subcellular Location
HanXRQr2_Chr02g0066401	*HaTGA01*	362	41,390.12	8.73	47.1	83.54	−0.493	Nucl
HanXRQr2_Chr02g0080541	*HaTGA02*	499	56,214.97	6.75	60.01	73.51	−0.617	Nucl
HanXRQr2_Chr04g0158781	*HaTGA03*	359	41,090.07	8.89	52.57	82.4	−0.437	Nucl
HanXRQr2_Chr05g0236811	*HaTGA04*	328	37,714.8	6.77	52.46	88.63	−0.51	Nucl
HanXRQr2_Chr06g0271641	*HaTGA05*	328	36,462.02	8.63	61.5	81.62	−0.53	Nucl
HanXRQr2_Chr09g0385991	*HaTGA06*	359	40,422	5.96	43.45	89.75	−0.416	Nucl
HanXRQr2_Chr10g0432971	*HaTGA07*	508	56,579.34	6.34	62.6	74.39	−0.627	Nucl
HanXRQr2_Chr11g0467021	*HaTGA08*	365	41,616.81	8.92	53	89.32	−0.38	Nucl
HanXRQr2_Chr12g0554701	*HaTGA09*	326	36,480.08	8.63	60.54	81.23	−0.538	Nucl
HanXRQr2_Chr13g0568161	*HaTGA10*	417	46,324.83	8.77	54.27	73.53	−0.618	Nucl
HanXRQr2_Chr13g0581141	*HaTGA11*	512	56,907.53	6.53	56.58	73.55	−0.578	Nucl
HanXRQr2_Chr14g0629161	*HaTGA12*	328	36,392.94	7.9	59.46	83.38	−0.492	Nucl
HanXRQr2_Chr15g0718871	*HaTGA13*	364	40,896.45	5.7	41.62	86.13	−0.387	Nucl
HanXRQr2_Chr17g0803501	*HaTGA14*	466	51,558.8	6.21	48.53	79.81	−0.461	Nucl

## Data Availability

All the other data sets supporting the conclusions of this article are included within the article and its Appendix A.

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
