# Peer review of "Genome-Wide Identification and Expression Analysis of TGA Family Genes Associated with Abiotic Stress in Sunflowers (Helianthus annuus L.)"

_ijms, 2024, doi:10.3390/ijms25074097_

Round 1

Reviewer 1 Report

Comments and Suggestions for Authors

Dear Editor,

I have completed my review of the manuscript entitled "Genome-Wide Identification and Expression Analysis of TGA Family Genes Associated with Abiotic Stress in Sunflowers (Helianthus annuus L.)", submitted for publication in the International Journal of Molecular Sciences. This manuscript presents a comprehensive analysis of the TGA gene family in sunflowers, highlighting its potential role in abiotic stress tolerance. The topic is of significant interest to the field of plant molecular biology and stress physiology, and the study offers valuable insights into the genetic and molecular mechanisms underpinning stress responses in sunflowers.

While I appreciate the efforts of the authors and recognize the potential impact of their work, I have identified several areas where improvements are necessary to enhance the clarity, rigor, and significance of the manuscript. Below, I provide specific scientific and grammatical comments for consideration:

Scientific Comments

  1. Methodological Details: The criteria and methodology for identifying TGA genes in sunflowers should be described more explicitly. Clarifying the BLAST query parameters and redundancy removal process will improve the manuscript's reproducibility and rigor.

  2. Gene Distribution and Structural Diversity: The authors are encouraged to discuss the potential functional implications of the uneven distribution of TGA genes across chromosomes and the variability in their exon-intron structures. How might these genetic features influence the functional diversity and stress response capabilities of TGA genes in sunflowers?

  3. Evolutionary Insights from Collinearity Analysis: The manuscript would benefit from a deeper exploration of the evolutionary significance of gene duplication events identified in the collinearity analysis. Discussing how these events might contribute to the functional diversification of TGA genes in abiotic stress responses would enrich the narrative.

  4. Functional Implications of Phylogenetic Grouping: The categorization of TGA proteins into four main branches suggests evolutionary divergences with potential functional implications. The manuscript should explore these implications further, particularly in the context of abiotic stress adaptation.

  5. Regulatory Networks Inferred from Cis-acting Elements: The analysis of cis-acting elements in the promoters of TGA genes unveils potential regulatory networks. A discussion on how these networks might orchestrate the plant's response to abiotic stresses and hormonal signals would provide a more comprehensive understanding of TGA gene functions.

  6. Interpretation of Differential Gene Expression: The differential expression of HaTGA genes under various stress conditions is a pivotal finding. The manuscript should discuss the adaptive significance of these expression patterns, considering sunflower's resilience to abiotic stresses.

Grammatical Comments

  1. Terminology and Abbreviations: Ensure consistency in the use of terms and abbreviations throughout the manuscript. This includes the standardized abbreviation of transcription factors as "TF" after its first mention.

  2. Simplification of Complex Sentences: Some sentences in the Results section are overly complex. Simplifying these sentences will improve readability and comprehension.

  3. Proofreading for Typos and Errors: A thorough proofreading of the manuscript is recommended to correct minor typographical errors and grammatical inconsistencies, ensuring a polished and professional presentation.

Thank you for the opportunity to review this work. I look forward to seeing the revised manuscript.

Sincerely,

Comments on the Quality of English Language

The manuscript should be checked by a native English person.

Reviewer 2 Report

Comments and Suggestions for Authors

This work is a valuable contribution in genome-wide identification of TGA transcription factors in sunflower, with an attempt to study the role of these factors in hormonal treatment, flooding and salt stress responses in leaves. However, a major revision is necessary in order to be published.

Introduction – the first paragraph (especially lines 33-40), although interesting and illustrating the stress tolerance of sunflower, is not at all related to TGA transcription factors. Many facts concerning TGA role are cited mechanistically, without apparent links to the purpose of the study.

MMs - 4.6. Plant material and treatments should be described in more details. In what conditions plants were grown, how the treatments were applied, why leaves were chosen for analysis?

Results. I could not find any figures in the text and there should be at least seven, according to the description of results.

Line 85 – “of genes. We performed…” – should be in one sentence. The same for line 147 –“ of HaTGA. As depicted

In the text there are some repetitions, such as: Line 82 – “HaTGA02 contains the largest contained the largest…”, line 340 – “including including”

Discussion – more about the biological role of TGA factors could be said

Comments on the Quality of English Language

needs improvement

Round 2

Reviewer 1 Report

Comments and Suggestions for Authors

Dear Editor.

All comments have been addressed.

The manuscript can be accepted now.

Reviewer 2 Report

Comments and Suggestions for Authors

The revised version of the manuscript is substantially improved. The mising figures are presented in the text.